



# A sea ice deformation and rotation rates dataset (2017-2023) from the Environment and Climate Change Canada Automated Sea Ice Tracking System (ECCC-ASITS)

Mathieu Plante[1], Jean-François Lemieux[1], L. Bruno Tremblay[2], Amélie Bouchat[2], Damien Ringeisen[2,6], Philippe Blain[3], Stephen Howell[4], Mike Brady[4], Alexander S. Komarov[5], Béatrice Duval[2], and Lekima Yakuden[2]

[1]Recherche en prévision numérique environnementale, Environnement et Changement Climatique Canada, Dorval, Québec, Canada.
[2]Department of Atmospheric and Oceanic Sciences, McGill University, Montréal, Québec, Canada
[3]Service Météorologique Canadien, Environnement et Changement Climatique Canada, Dorval, Québec, Canada.
[4]Climate Research Division, Environment and Climate Change Canada, Toronto, Canada.
[5]Meteorological Research Division, Environment and Climate Change Canada, Ottawa, Canada.
[6]NASA Goddard Institute for Space Studies and Columbia University, New York, USA.

**Correspondence:** Mathieu Plante (mathieu.plante@ec.gc.ca)

**Abstract.** Sea ice forms a thin but horizontally extensive boundary between the ocean and the atmosphere, with a complex crust-like dynamics characterized by intermittent sea ice deformations. The heterogeneity and localisation of these sea ice deformations are important characteristics of the sea ice cover that can be used to evaluate the performance of dynamical sea-ice models against observations across multiple spatial and temporal scales. Here, we present a new pan-Arctic sea-ice deformation and rotation rates (SIDRR) dataset derived from the RADARSAT Constellation Mission (RCM) and Sentinel-1 (S1) synthetic aperture radar (SAR) imagery from 01 September 2017 to 31 August 2023. The SIDRR estimates are derived from contour integrals of triangulated ice motion data, obtained from the Environment and Climate Change Canada automated sea ice tracking system (ECCC-ASITS). The SIDRR dataset is not regularised, and consist in stacked data from multiple SAR images computed on a range of temporal (0.5 - 6 days) and spatial (4-10 km) scales. It covers the entire Arctic Ocean and all peripheral seas except the Okhotsk Sea. Uncertainties associated with the propagation of tracking errors on the deformation values are included. We show that rectangular patterns of deformation features are visible when the sampled deformation rates are lower than the propagation error. This limits the meaningful information the can be extracted in areas with low SIDRR values, but allows for the characterisation of SIDRR in Linear Kinematic Features. The spatial coverage and range of resolutions of the SIDRR dataset provides an interesting opportunity to investigate regional and seasonal variability of sea-ice deformation statistics across scales, and can be used to determine metrics for model evaluation.

## 1 Introduction

As the perennial Arctic sea ice is declining, an increasing number of vessels are seasonally seen navigating the polar routes (Pizzolato et al., 2014, 2016; Dawson et al., 2018). Navigation in ice-infested waters however remains hazardous, leaving





vessels without or with limited ice-breaking capability vulnerable to the changing sea ice conditions (Mudryk et al., 2021; Chen et al., 2022). In particular, any changes to the sea ice drift may also result in rapidly building sea ice pressure and far-reaching material deformations such as fracturing, ridging and lead opening. The type, timing and location of these sea ice deformations can either ease or impede navigation and safety operations, but remain difficult to predict due to the complex dynamics of sea ice as a thin layer of solid material subjected to large-scale surface forces (e.g. winds, tides and ocean currents). This difficulty can in large part be attributed to the multi-scale character of this dynamics (Kwok, 2001; Lindsay and Stern, 2003; Marsan et al., 2004; Rampal et al., 2019) in a region with scarce observations, complicating model development and validation (Bouchat et al., 2022; Hutter et al., 2022).

Observations of sea ice deformation need to be derived from arrays of motion vectors, measured either in situ from drifting buoys (Heil et al., 1998; Hutchings et al., 2011, 2012; Itkin et al., 2017; Lei et al., 2020; Womack et al., 2024) or remotely by applying feature recognition and tracking methods on satellite imagery (Kwok, 2001; Komarov and Barber, 2014; Linow and Dierking, 2017; Korosov and Rampal, 2017; von Albedyll et al., 2024) or ship radar (Oikkonen et al., 2017). On the one hand, the in situ observations have the advantage of producing high precision data with frequent temporal sampling, but their point-data character usually makes for limited spatial coverage. On the other hand, remote sensing observations gather a large amount of data with multi-scale information and coverage, but yield lower precision data and limited temporal sampling. An advantage of the remote sensing approaches is that they can be used to produce pan-Arctic sea ice motion fields (Lavergne et al., 2010; Tschudi et al., 2020; Tian et al., 2022; Lavergne and Down, 2023), and thus define sea ice motion seasonality, inter-annual variability and climatological trends. Gridded sea ice motion products however tend to involve filtering and post-processing methods that are not designed or appropriate for computing sea ice deformations. For this reason, remote-sensing-derived sea ice deformations are usually produced independently of gridded sea ice motion products (e.g., Kwok, 2001; Bouillon and Rampal, 2015; von Albedyll et al., 2021).

In particular, the RADARSAT Geophysical Processing System (RGPS) (Kwok et al., 1998) was purposefully designed for the study of sea ice dynamics and was seminal for the establishment of large (Kwok et al., 2008) and multi-scale (Marsan et al., 2004) sea ice deformation properties. This dataset remains to this day the most common sea ice deformation reference for model development and validation (e.g., Bouchat and Tremblay, 2017; Spreen et al., 2017; Rampal et al., 2019; Hutter et al., 2018; Bouchat et al., 2022; Ringeisen et al., 2022). The RGPS dataset corresponds to a list of Lagrangian tracks in the central Arctic produced by applying feature tracking methods on synthetic aperture radar (SAR) imagery (Kwok et al., 1998). Each winter from 1997 to 2008, the months-long trajectories are initialized from SAR features organised on a regular 10-km-resolution grid, then tracked for many months until spring (Kwok et al., 1998; Kwok and Cunningham, 2002). The trajectories were later used to derive the RGPS sea ice deformations (Kwok, 2001), from which a number of multi-scale properties of sea ice dynamics has been established such as the power law of its observed Probability Distribution Function (PDF) and spatio-temporal scaling properties (Bouchat and Tremblay, 2017; Rampal et al., 2019; Bouchat and Tremblay, 2020; Hutter and Losch, 2020). Currently, these diagnostics are the main source of validation for rheological models (e.g., in Ólason et al., 2022; Bouchat et al., 2022; Hutter et al., 2022), despite the widening gap between this reference period and the current state of the sea ice cover.





In recent years, our ability to observe pan-Arctic sea ice motion has been improved by the deployment of several satellites equipped with SAR and with high pass frequency in the Arctic: the Sentinel 1 mission (S1, 2 satellites) and the Radarsat Constellation Mission (RCM, 3 satellites). Sophisticated SAR feature tracking algorithms have also been developed to efficiently process large numbers of SAR images (Komarov and Barber, 2014; Howell et al., 2022) and used to study regional sea ice dynamics (Babb et al., 2021; Moore et al., 2021). Together, this paved the way for the development of the first operational sea ice motion product at Environment and Climate Change Canada (ECCC), based on an automated sea ice tracking system (ASITS Howell et al., 2022). Designed to optimise the number of images processed to cover the entire Arctic, the ECCC-ASITS uses a combination of S1 and RCM SAR images to determine the sea ice motion at a 25 and 6.25 km resolution. The ECCC RCM/S1 sea ice motion product is available from 2020 to present (Brady and Howell, 2021).

Here, we present a new pan-Arctic Sea-Ice Deformation and Rotation Rates (SIDRR) dataset based on S1 and RCM SAR imagery and derived from the raw (non-gridded) ECCC-ASITS outputs. The SIDRR dataset covers the entire Arctic at variable time and space resolutions depending on the image acquisition and data processing but in the ranges of 2-10 km and 0.5-6 days. The data organisation, its coverage and uncertainties are presented, and limitations inherent to the use of the operational ASITS are discussed.

The manuscript is organized as follows. Input data from the ECCC-ASITS used to produce the SIDRR dataset is described in Sect. 2. The production algorithm, SIDRR computation and associated uncertainties are presented in Sect. 3. Characteristics of the dataset are detailed in Sect. 4, including the data format, the range of spatio-temporal scales and coverage. A discussion on data validation and uncertainties is provided in Sect. 5, with conclusions summarized in Sect. 6.

## 2   Input Data: ECCC-ASITS Sea Ice Motion

The ECCC-ASITS was developed to routinely generate pan-Arctic sea ice motion fields by processing an optimised number of SAR images to cover entire Arctic within a limited time frame. The SAR images are taken from the S1 and RCM satellites, with images from each mission processed independently in different algorithm streams. The S1 mission includes two satellites (S1A and S1B) launched in 2014, of which the S1A is to this day still operational but S1B stopped transmitting data in March 2021. The RCM includes three satellites launched in June 2019. The SIDRR dataset S1-stream thus covers the entire processed period (2017-2023), while the RCM-stream is added from January 29, 2020 onward.

The ECCC-ASITS proceeds as follows. First, the Arctic is divided into 400 km x 400 km sectors that are used to optimise the number of SAR images necessary to cover the entire Arctic. For each sector, a stack of images is created by selecting images that have at least a $\sim$30 % area overlap with the sector. The images within that stack are then paired if their area are overlapping by at least 32 000 km$^2$. Each initial image is only paired once–with the first match satisfying these criteria–before being removed from the stack. The algorithm of Komarov and Barber (2014) is applied on each selected pair to generate the sea ice motion vectors, using a tracking resolution of 200 m and tracked feature spacing of $\sim$8 km. Results from each pair of SAR images are output in a text file listing all identified tracked features with their start location and displacement in X-Y pixel coordinates, latitudes, longitudes, and a measure of confidence level in the feature tracking. Here, we use the raw sea ice





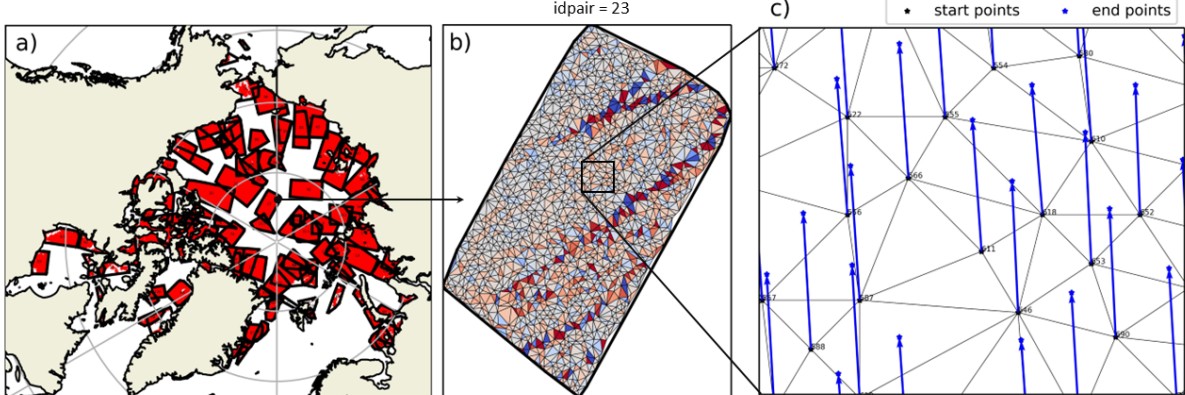

**Figure 1.** Organisation of the dataset: a) tracked features (red) in the file "SIDRR_20210212.nc", from the stacked SAR image pairs (black contours), b) Sea ice divergence calculated on the triangulated array from idpair = 23, and c) zoom on triangular arrays from idpair = 23, with the start positions (black dots), drift (blue arrows) and the end position(blue dots) of the tracked features. The vertex ID numbers are indicated in black.

motion vectors directly from the tracking algorithm to generate the new sea ice deformation product (i.e., we do not use the ECCC-ASITS SIM gridded datasets, which are produced by aggregating and post-processing the SIM vectors).

## 3 Methods

A Python algorithm was developed to read the raw ECCC-ASITS output files, compute the sea ice deformations and uncertainties, and stack results in daily netCDF output files. As in the ECCC-ASITS, each image pair is processed individually, organizing its tracked feature start locations into a list of triangular arrays using Delaunay triangulation (Fig. 1). Triangles that are too thin (with angles < 10°) or too big (area A>400 km$^2$) are discarded. The sea ice drift, motion and deformations are computed for each triangle based on the end location of the tracked vertexes (Fig. 1c). Note that as the SIDRR are computed from the tracked feature locations, they differ from results computed from the post-processed RCM/S1 sea ice motion product.

### 3.1 Sea ice deformation computation

For each triangle, the sea ice deformation is computed using a line integral method (Bouchat and Tremblay, 2020; Bouchat et al., 2022) based on the velocity and start location of the tracked feature at each vertex. The area $A$ of the triangle in the first image is first defined as

$$A = \frac{1}{2}\sum_{i=1}^{N}(x_i^t y_{i+1}^t - x_{i+1}^t y_i^t), \tag{1}$$

where $t$ represent the time level (acquisition time) of the first image, $N$ is the number of vertex in the array (here 3), and $x$ and $y$ indicate the position of the vertex $i$ ($i$ increasing counterclockwise, using $N+1=1$) in the x-axis and y-axis respectively.

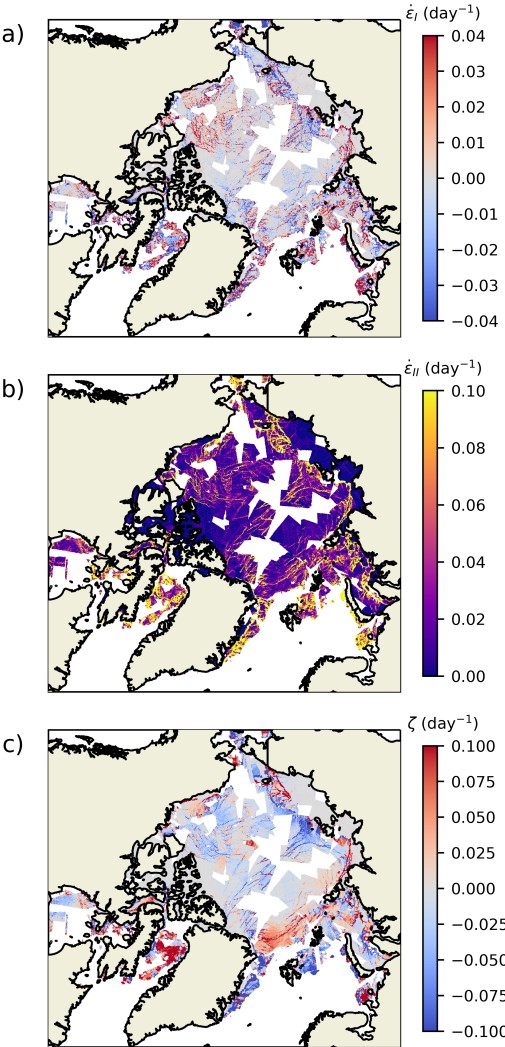

**Figure 2.** Example of sea ice divergence (a), shear (b) and vorticity (c) on 2021-02-12.

The sea ice motion components at each vertex are defined as:

$$u = \frac{x^{t+\Delta t} - x^t}{\Delta t}, \qquad\qquad v = \frac{y^{t+\Delta t} - y^t}{\Delta t}, \qquad\qquad (2)$$

where $u$ and $v$ represent the motion of the tracked features in the x and y direction respectively and $\Delta t$ indicates the acquisition time difference between the paired images.



The strain rates are then given by:

$$u_x = \frac{1}{A} \sum_{i=1}^{N} (u_{i+1} + u_i)(y_{i+1}^t - y_i^t), \qquad (3)$$

$$u_y = -\frac{1}{A} \sum_{i=1}^{N} (u_{i+1} + u_i)(x_{i+1}^t - x_i^t), \qquad (4)$$

$$v_x = \frac{1}{A} \sum_{i=1}^{N} (v_{i+1} + v_i)(y_{i+1}^t - y_i^t), \qquad (5)$$

$$v_y = -\frac{1}{A} \sum_{i=1}^{N} (v_{i+1} + v_i)(x_{i+1}^t - x_i^t), \qquad (6)$$

where the $x$ and $y$ subscripts represent derivatives in the x and y directions respectively.

The normal deformation rates $\dot{\epsilon}_\mathrm{I}$, shear deformation rates $\dot{\epsilon}_\mathrm{II}$ and vorticity $\dot{\Omega}$ are written as :

$$\dot{\epsilon}_\mathrm{I} = u_x + v_y, \qquad (7)$$

$$\dot{\epsilon}_\mathrm{II} = \sqrt{(u_x - v_y)^2 + (u_y + v_x)^2}, \qquad (8)$$

$$\dot{\Omega} = v_x - u_y. \qquad (9)$$

The total deformation rates is finally obtained from the strain rate invariants:

$$\dot{\epsilon}_\mathrm{tot}^2 = \dot{\epsilon}_\mathrm{I}^2 + \dot{\epsilon}_\mathrm{II}^2. \qquad (10)$$

To provide an example, the divergence deformation rates, shear deformation rates and rotation rates on Februray 12th, 2021 are presented in Fig. 2. The computed sea ice deformation are in the range of -1.0 to 1.0 day$^{-1}$ for divergence and rotation, and from 0.0 to 1.0 day$^{-1}$ for shear. The SIDRR data capture well the organization of the sea ice dynamics into localised lines of large deformation (known as linear kinematic features, or LKFs) and low deformation elsewhere. Large deformations are also seen in the marginal ice zone, where the rheology plays a lesser role due to the fragmented state of sea ice.

### 3.2 Tracking uncertainty

The computed sea ice deformations have uncertainties associated with the tracking errors (propagation error) and with the small (3) number of points used to perform the contour integrals (boundary errors). These errors represent important limitations and need to be taken into account when producing sea ice deformation characteristics. While the boundary errors are difficult to quantify and need to be treated as part of post processing by the users (Lindsay and Stern, 2003; Bouillon and Rampal, 2015; Griebel and Dierking, 2018), the propagation error is defined using propagation of uncertainties principles (Hutchings et al., 2011; Bouchat and Tremblay, 2020; Dierking et al., 2020). As such, in the SIDRR dataset, each computed sea ice deformation is assigned to a signal-to-noise value that can be used to filter the data according to the propagation error.



**Table 1.** Description of dataset variables and dimensions

| Variable | Dim. | Description | Units | Variable | Dim. | Description | Units |
|---|---|---|---|---|---|---|---|
| - | n | triangle ID number | - | end_lon1 | n | End longitude, tracked vertex 1 | ° E |
| idpair | n | SAR scene pair ID number | - | end_lon2 | n | End longitude, tracked vertex 2 | ° E |
| ids1 | n | ID of triangle vertex 1 | - | end_lon3 | n | End longitude, tracked vertex 3 | ° E |
| ids2 | n | ID of triangle vertex 2 | - | A | n | Start triangle Area | m$^2$ |
| ids3 | n | ID of triangle vertex 3 | - | dudx | n | x-direction divergence rate | day$^{-1}$ |
| start_time | n | Acquisition time, SAR image #1 | hours | dvdy | n | y-direction divergence rate | day$^{-1}$ |
| end_time | n | Acquisition time, SAR image #2 | hours | dvdx | n | x-direction shear rate | day$^{-1}$ |
| start_lat1 | n | Start latitude, tracked vertex 1 | ° N | dudy | n | y-direction shear rate | day$^{-1}$ |
| start_lat2 | n | Start latitude, tracked vertex 2 | ° N | div | n | Divergent deformation rate ($\dot{\epsilon}_\mathrm{I}$) | day$^{-1}$ |
| start_lat3 | n | Start latitude, tracked vertex 3 | ° N | shr | n | Shear deformation rate ($\dot{\epsilon}_\mathrm{II}$) | day$^{-1}$ |
| start_lon1 | n | Start longitude, tracked vertex 1 | ° E | vrt | n | vorticity ($\dot{\Omega}$) | day$^{-1}$ |
| start_lon2 | n | Start longitude, tracked vertex 2 | ° E | err_div | n | Propagation error on $\dot{\epsilon}_\mathrm{I}$ | day$^{-1}$ |
| start_lon3 | n | Start longitude, tracked vertex 3 | ° E | err_shr | n | Propagation error on $\dot{\epsilon}_\mathrm{II}$ | day$^{-1}$ |
| end_lat1 | n | End latitude, tracked vertex 1 | ° N | err_vrt | n | Propagation error on $\dot{\Omega}$ | day$^{-1}$ |
| end_lat2 | n | End latitude, tracked vertex 2 | ° N | errA | n | Propagation error on $A$ | m$^2$ |
| end_lat3 | n | End latitude, tracked vertex 3 | ° N | s2n | n | signal to noise ratio | - |

Assuming a positional error $\sigma_x$, the propagated error on the computed area ($\sigma_A$) and velocity ($\sigma_u$) are obtained from:

$$\sigma_A^2 = \frac{1}{4}\sum_{i=1}^{N}\left[(x_{i+1}-x_{i-1})^2 + (y_{i+1}-y_{i-1})^2\right]\sigma_x^2, \tag{11}$$

$$\sigma_u^2 = \frac{\sigma_x^2}{\Delta t^2}. \tag{12}$$

The propagated error on the computed deformation components is then expressed as (here showing only one component and neglecting the geolocalization error, see Bouchat and Tremblay, 2020, for more details):

$$\sigma_{u_x}^2 = u_x^2\left(\frac{\sigma_A}{A}\right)^2 + \sum_{i=1}^{N}\left(\frac{y_{i+1}-y_{i-1}}{2A}\right)^2\sigma_u^2$$
$$+ \sum_{i=1}^{N}\left(\frac{u_{i+1}-u_{i-1}}{2A}\right)^2\sigma_x^2, \tag{13}$$



where $\sigma_{u_x}$ represent the error on $u_x$. The errors on the strain rate invariants and rotation rates are expressed in terms of the errors on each components, as:

$$\sigma_{\dot{\epsilon}_\mathrm{I}}^2 = \sigma_{u_x}^2 + \sigma_{v_y}^2, \tag{14}$$

$$\sigma_{\dot{\epsilon}_\mathrm{II}}^2 = \left(\frac{u_x - v_y}{\dot{\epsilon}_\mathrm{II}}\right)^2 \sigma_{\dot{\epsilon}_\mathrm{II}}^2 + \left(\frac{u_y + v_x}{\dot{\epsilon}_\mathrm{II}}\right)^2 (\sigma_{u_y}^2 + \sigma_{v_x}^2), \tag{15}$$

$$\sigma_{\dot{\Omega}}^2 = \sigma_{u_y}^2 + \sigma_{v_x}^2, \tag{16}$$

and the error on the total deformation rate is:

$$\sigma_{\dot{\epsilon}_\mathrm{tot}}^2 = \frac{\dot{\epsilon}_\mathrm{I}^2 \sigma_{\dot{\epsilon}_\mathrm{I}}^2 + \dot{\epsilon}_\mathrm{II}^2 \sigma_{\dot{\epsilon}_\mathrm{II}}^2}{\dot{\epsilon}_\mathrm{tot}^2}. \tag{17}$$

The signal-to-noise ratio $s$ is then defined as the relative magnitude of the total deformation rates with respect to the total deformation error:

$$s = \frac{\dot{\epsilon}_\mathrm{tot}}{\sigma_{\dot{\epsilon}_\mathrm{tot}}} = \frac{\dot{\epsilon}_\mathrm{tot}^2}{\sqrt{\dot{\epsilon}_\mathrm{I}^2 \sigma_{\dot{\epsilon}_\mathrm{I}}^2 + \dot{\epsilon}_\mathrm{II}^2 \sigma_{\dot{\epsilon}_\mathrm{II}}^2}}. \tag{18}$$

## 4 Dataset Description

### 4.1 Format

The SIDRR data is stored in daily netcdf files using a "SIDRR_YYYYMMDD.nc" nomenclature. Each file contains the data from all paired SAR images with an initial (first image) acquisition time within the given date. The recorded variables are listed in Table 1 and have the shape of vectors of length "n" corresponding to the total number of triangles stacked in the netcdf files. The dataset includes all the necessary information to visualise or post-process the SIDRR data. ID numbers are also provided to identify each SAR image pair ("idpairs"), triangle ("n") and tracked feature ("ids1","ids2","ids3", one for each vertex). The tracked feature IDs are attributed independently (i.e., they recur) for each image pair.

The SIDRR data has not been regularised and corresponds to a patchwork of data from different SAR scenes computed over a range of spatial-temporal scales (see section 4.2 below). As such, there are large areas without data between the processes scenes, or with overlaps (see for instance in Fig. 1a). This differs from the RGPS dataset in which the tracked features are initially organized on a 10-km resolution rectilinear grid and then tracked for the entire winter, allowing for a nearly contiguous mapping of sea ice deformations in the central Arctic (Kwok et al., 2008; Bouchat and Tremblay, 2020). Here, the post-processing steps necessary to produce a contiguous (regularized) pan-Arctic mapping of sea ice deformations remains to be established, and is left for future work. This allows using this SIDRR dataset to test different post-processing approaches and their impact on diagnostics used for model validation.

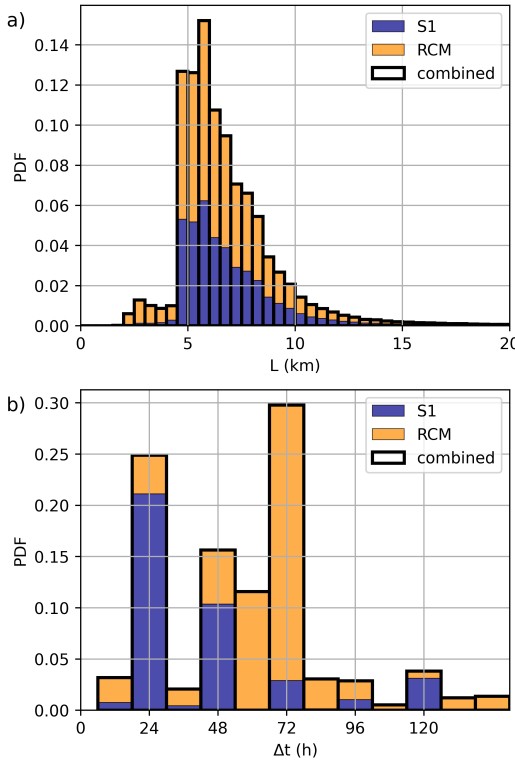

**Figure 3.** Distribution of the spatial (a) and temporal (b) scales of the computed deformations, in the period with both S1 and RCM data (2020-02-01 to 2023-08-31). Colors indicate the contribution from data in the S1 and RCM streams.

## 4.2 Spatio-temporal caracteristics

### 4.2.1 Scale

The SIDRR are computed from triangular arrays with variable sizes; the effective radius ($L = \sqrt{A}$) of each triangle in the dataset are distributed around a mean of $\sim 6.7\,\mathrm{km}$, but range from a minimum of $2\,\mathrm{km}$ and a maximum of $\sim 15\,\mathrm{km}$ (Fig. 3a). This distribution is coherent with the target feature-spacing resolution in the ECCC-ASITS ($\sim 8\,\mathrm{km}$), and variations associated with the exact location (or absence) of identifiable SAR texture features. Triangles with effective radius larger than $20\,\mathrm{km}$ are discarded to avoid including data from erroneous arrays formed with lone tracked points along the coasts or near the marginal ice zones. Note that the distributions are nearly identical for data computed from S1 and RCM images, unsurprisingly given that the images are processed using the same algorithm, although in different streams.

The SIDRR data are also computed at different temporal scales (Fig. 3b), depending on the time difference between the acquisition of the paired SAR images. These time intervals depend on the method used in the ECCC-ASITS to pair the SAR images (Howell et al., 2022); this is made sequentially by selecting the first image satisfying some set of criteria (e.g.,

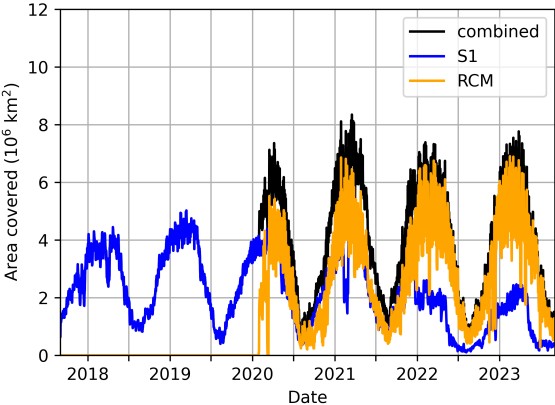

**Figure 4.** Time series of the area with valid deformation data from 01 September 2017 to 31 August 2023, for the S1 data (blue), the RCM data (orange) and both combined (black).

minimum time, minimum overlapping area). As the S1 and RCM images are processed in different streams, their temporal scale distributions present differences associated with their respective frequency of satellite passes. Most of the S1 image pairs present 24 h intervals, with many sub-daily and some at 48, 72 and 96 h intervals. The RCM image pairs mostly present 60 and 72 h intervals. These differences largely affect the magnitude of the computed sea ice deformations, and need to be taken into account when producing pan-Arctic SIDRR statistics.

### 4.2.2 Coverage

The total area covered daily in the SIDRR dataset has a strong seasonal cycle, in part due to the seasonality in the sea ice cover itself but also to the difficult identification of SAR features in summer, reducing the success rate of the tracking algorithm. Winter season (January to March) features a maximum coverage of $\sim$4 million km$^2$ before 2020, climbing to $\sim$7 million km$^2$ once RCM data is included. This is about $\sim$50 % of the pan-Arctic maximum sea ice extent (Fetterer et al., 2017). In summer (from July to September), the SIDRR dataset area coverage falls to $\sim$1.2 million km$^2$ (about $\sim$20 % of the NSIDC minimum extent). Including the RCM data thus nearly doubles the coverage of the dataset. Note that this represent a significant improvement compared to the RGPS coverage, which was limited to the central Arctic.

Despite the numerous gaps between SAR images in specific daily files, the percent daily coverage in winter is high (>50%) over most of the Arctic basin, nearing 100% days covered in some areas once the RCM data is added (from 2020 on-ward, see Fig. 5-a-c). The local temporal sampling is thus nearly daily over the entire Arctic, including all peripheral seas except the Okhotsk Sea, which is not covered by the ASITS. This demonstrates the benefits of including the RCM images: while the coverage from S1 images is generally good in the high Arctic, the RCM data shows a smaller North Pole blind spot and is more evenly distributed, raising the coverage in the peripheral seas. In summer, near-daily coverage is seen in the high Arctic (Northward of 80°N), but is low elsewhere where marginal ice zone dynamics dominates (Fig. 5-d-f).




**Figure 5.** Percent daily coverage from the S1/RCM sea ice deformation rates dataset in winter (from 01 January to 31 March) 2021 (a-c) and summer (01 July to 30 September) 2021 (d-f), using data from both the S1 and RCM streams (a,d), only the S1 stream (b,e) and only the RCM stream (c,f).

Note that the presence of square regions with higher coverage are associated with the ASITS method for optimising the number of SAR scenes to cover the entire Arctic, leaving a tendency for gaps at the border of each selection areas (see Fig. 6 in Howell et al., 2022). Increasing the number of processed RCM images can reduce this uneven sampling effect, but would need to be performed outside of the operational ASITS framework.

## 5  Validation and uncertainties

Validation of the SIDRR features is complicated by 1. the multi-scale nature of sea ice deformations, making for difficult interpretation of side-by-side comparisons, 2. the lack of independent sea ice motion product covering the same period at



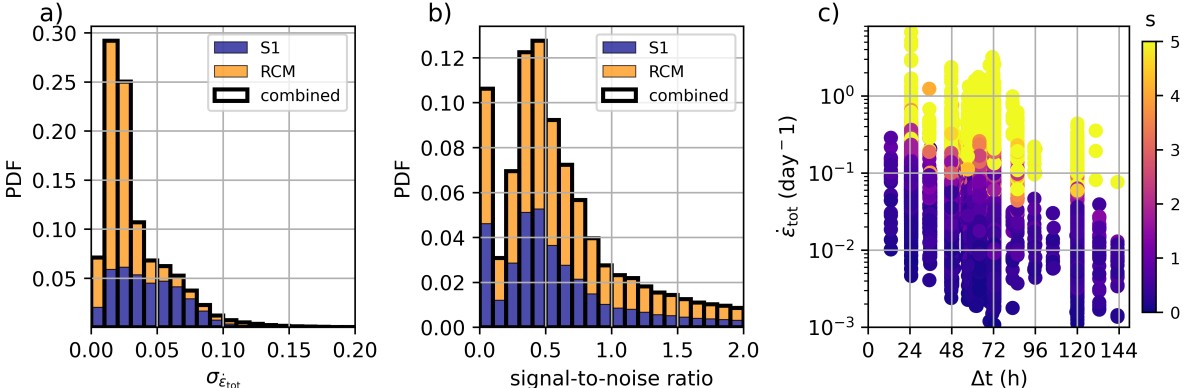

**Figure 6.** Distribution (combining the S1 and RCM stream) of total deformation rate propagation errors (a), distribution of the data signal-to-noise ratio (b) and scatter of the total sea ice deformation rates (c) as function of the computation time scale (x-axis) and the signal-to-noise ratio (in color), in the period with both RCM and S1 data (01 February 2020 to 31 August 2023. Colors in the distributions (a,b) indicate the contribution from data from the S1 and RCM streams.

similar resolution, and 3. the yet unknown impact of post-processing methods (part of producing gridded sea ice motion products) on sea ice deformation values. These difficulties are common to any sea ice deformation products, and the SIDRR
dataset is published in part to mitigate these difficulties by providing an independent yet flexible dataset that can be used by the sea ice dynamics community.

Nonetheless, the ECCC-ASITS sea ice motion themselves have been thoroughly validated and shown to have good correspondence against other products (see Howell et al., 2022, for details). Here, we use the ECCC-ASITS sea ice motion uncertainties–and their propagation into SIDRR errors–as a form of validation, and discuss the limitations they represent in
terms of the resolved features. Further validation in the form of comparisons with other sea ice deformation measurements is however planned for future work and will be conducted as part of in depth investigations of the SIDRR characteristics (power law, spatio-temporal scaling, LKFs) and their sensitivity to post-processing methods.

The sea ice displacement data from the ASITS was shown to have a (pan-Arctic) root mean square error (RMSE) of 2.78 km and mean deviation (MD) of 0.40 km against buoys. These values are coherent with the tracking being performed on a pixel
resolution (tracking error $\sigma_x$) of 200 m. Based on the propagation of this tracking error on the computed SIDRR values, the precision of the SIDRR data in is the range of $\sim$0-0.1 day$^{-1}$ (Fig. 6a), depending on the spatio-temporal scale of the computation. Much of the sea ice deformations display small signal-to-noise values (s<1, Fig. 6b). The data computed from RCM images present lower errors due to the predominance of longer (72 h) acquisition time difference. This uncertainty is quite large compared to other studies: in the RGPS data, the propagation of the tracking error on a computed area was estimated at
$\sim$1.4 % in the case of 100 km$^2$ quadrangle cells, leading to a sea ice deformation uncertainty of 0.005 day$^{-1}$ if computed over a 3 day period.



The larger SIDRR uncertainty in the present dataset is a consequence of making an opportunistic use of the ECCC-ASITS algorithm, which was designed to produce pan-Arctic sea ice motion maps within an operational time frame, balancing positional precision with computational costs. Precisely, compared to RGPS, our sea ice deformations has larger propagation error due to: (1) the tracking resolution of 200 m in the ECCC-ASITS algorithm (vs. 100 m in RGPS), (2) smaller (often 24h) time scales (vs. 3 days in RGPS) and (3) the computation of deformations over triangular arrays (vs. a rectangle grid in RGPS). It is likely that this uncertainty can be lowered by applying post-processing and data aggregation methods, although at the cost of loosing the smaller-scale information. This will be explored in future work.

The analysis reveals that the SIDRR data is well split between highly uncertain values in low deformation areas and meaningful values in high-deformation areas. In particular, low deformation areas are prone to display rectangular artefact patterns where the nominal resolution of the feature tracking is not sufficient to capture the gradients in sea ice displacement (Fig. 7a,b). These artefacts appear when the drift distance presents step-like jumps corresponding to the tracking resolution ($\sigma_x = 200$ m in our case, see Fig. 7c), concentrating an originally smooth deformation field into localised lines. The squared signal-to-noise values ($s^2$) can be used for a proxy to differentiate these artefact from LKFs, with $s^2 > 1.0$ for LKFs and $s^2 < 1.0$ for artefacts (Fig. 7d).

Note that the analysis above does not consider uncertainties associated with the approximation of SIDRR from triangular arrays. These boundary errors are not related to the tracking uncertainty but are likely as significant (Lindsay and Stern, 2003), as attested by the noisy appearance of the divergence rate fields in the resolved LKFs (see. Fig. 1-b). Different filtering approaches (e.g. Bouillon and Rampal, 2015; Griebel and Dierking, 2018) can (and must) be used by the user as part of post-processing of the SIDRR dataset, and will also be explored in future work.

## 6 Conclusions

A sea ice deformation and rotation rates (SIDRR) dataset is constructed based on the sea ice motion vectors obtained from the Environment and Climate Change Canada automated sea ice tracking system (Howell et al., 2022, ECCC-ASITS). Developed for operational purposes, the ECCC-ASITS retrieves the sea ice drift from conjoined SAR images by applying the feature tracking algorithm of Komarov and Barber (2014) on S1 and RCM data. The SIDRR are computed using a line integral approach based on the triangulated start (and end) location of the ECCC-ASITS tracked features.

The SIDRR dataset is organised in daily netcdf files, each stacking the SIDRR information from all the conjoined SAR images with an initial acquisition time within the given date. To preserve data flexibility, the dataset is kept as "raw" as possible and remains highly irregular, each file containing a patchwork SIDRR data from multiple SAR images. The SIDRR are defined from ranges of temporal (0.5 to 6 days) and spatial (~2.0 to ~10.0 km) scales. Different post-processing methods will be tested in future work to determine their influence on spatio-temporal scaling properties and other metrics commonly used for sea ice modelling and forecasting applications.

The SIDRR dataset (2017–2023) offers a year-long pan-Arctic (except the Okhotsk Sea) coverage, at least at a weekly frequency with a significant portion presenting daily coverage. The inclusion of summer months and the marginal seas is a

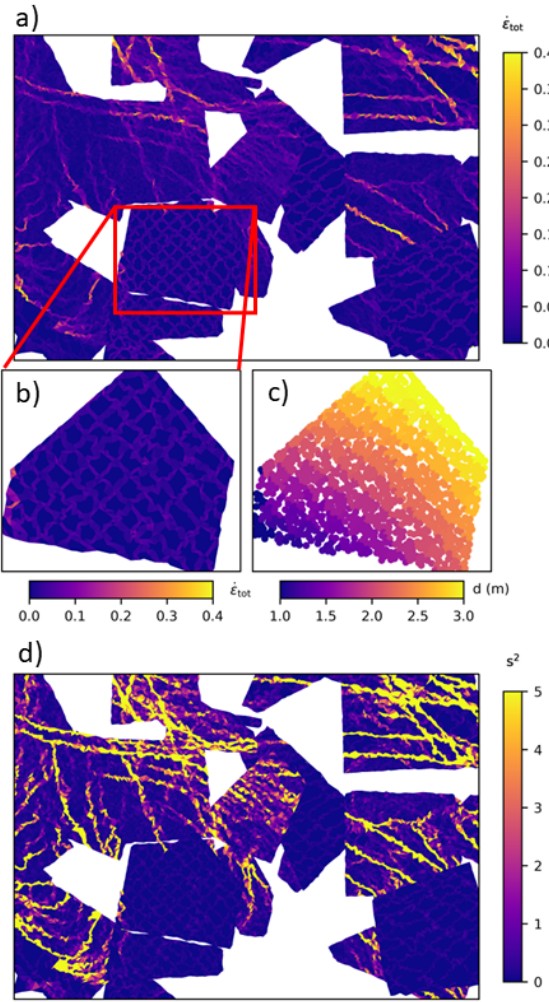

**Figure 7.** Example of artefacts associated with the feature tracking resolution not resolving gradients in sea ice displacement. Larger panels show the total sea ice deformation rates (a) and squared signal-to-noise ratio (d) from multiple SAR scene in the central Arctic close to the North Pole, on 06 March 2021. Small panels show a zoom on the total sea ice deformation rates in SAR image pair with ID = 252 (b) with the associated poorly resolved (step-like) displacement field (c). The squared signal-to-noise ratio in (d) effectively differentiate the artefacts from the LKFs.

significant improvement from the commonly used RGPS dataset (Kwok et al., 1998). The inclusion of RCM data from January 2020 onward significantly improves the spatio-temporal coverage, reaching ∼ 50 % and 20 % of the sea ice cover in winter and summer respectively. This extensive coverage represents an asset to investigate differences in sea ice dynamics between the recent (2017-2023) and RGPS periods (1997-2008), to establish pan-Arctic and regional SIDRR spatio-temporal characteristics and study their seasonality.

The SIDRR is shown to resolve localised zone of large deformations (or Linear Kinematic Features, LKFs) between areas of uncertain and low deformations. However, the relatively low feature-tracking resolution (200 m) yields large uncertainties in SIDRR (significantly larger than in the RGPS dataset). These uncertainties are shown to cause localised artefacts similar to LKFs. Signal-to-noise values are proven to be efficient in differentiating the certain high-deformation features in LKFs from these tracking-error artefacts. The extent at which these errors (along with boundary errors) can be reduced via post-processing

methods will assessed in future work.

    Much of the constraints in the current analysis come from the ECCC-ASITS being designed for operational purposes, with a need for computational efficiency. Note that despite these constraints, the operational character of the ECCC-ASITS remains beneficial with its ongoing production, making for likely yearly extensions of the present SIDRR dataset in the coming years. The constrains nonetheless leaves much potential for future improvements to the dataset by 1. the reduction of the tracking

errors, 2. increasing the resolution of the tracked features (increasing the SIDRR dataset resolution) and 3. increasing the number of processed SAR images. These improvements are currently under development and will allow for useful extensions of sea ice deformation observations from remote sensing towards the kilometer scale.

## 7   Code and data availability

The SIDRR dataset is available on Zenodo (Plante et al., 2024a). The Python code used to produce the SIDRR dataset is avail-

275 able on github at : https://github.com/McGill-sea-ice/ice-tracker-deformations. The Python scripts used for the data analysis and manuscript figures are available on Zenodo (Plante et al., 2024b).

*Author contributions.*   The ECCC-ASITS sea ice motion vector outputs are provided by MB, AK and SH, along with technical support and guidance. BD and LY developed the Python code to process the ECCC-ASITS data and compute the sea ice deformations, with contributions from AB, PB, JFL, MP, DR and BT. The data analysis and manuscript figures were coded and analysed by MP, AB, DR, LY and BD. MP

wrote the manuscript with contributions from AB, AK, SH, JFL, DR and BT.

*Competing interests.*   The authors declare that they have no conflict of interest.

*Acknowledgements.*   This work is a contribution to a Canadian Space Agency (CAS) Research Opportunity in Space Science (ROSS-Cycle III) grant #23SUESDEFO awarded to Tremblay.





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
