# Peer review of "A sea ice deformation and rotation rates dataset (2017-2023) from the Environment and Climate Change Canada Automated Sea Ice Tracking System (ECCC-ASITS)"

_Earth System Science Data, 2024_

## Author Comment (AC1)

**Answers to essd-2024-227 RC1**

October 9, 2024

Note:

- The referee comments are shown in black,
- The authors answers are shown in blue,
- *Quoted texts from the revised manuscript are shown in italic and in dark blue.*

The manuscript describes the methodology and the dataset of sea ice deformation and rotation derived from S1 and RCN-based sea ice drift. It is very well written and provides all the necessary information for users of the new dataset. Aspects of deformation computation and uncertainty estimates are covered in good detail. Artefacts appearing on ice deformation maps due to the coarse resolution of SAR data used for ice drift retrieval are well highlighted.

It is an excellent work, and I have only two minor comments on the manuscript's text. However, I have a major comment on the provided data. I recommend updating the dataset before the manuscript can be published, and I provide detailed instructions on how it can be done efficiently.

We thank you for this useful review. We agree with all your comments and address them below.

Minor comments on the text

Line 61. The phrase "combination of S1 and RCM SAR" reads as ice drift was derived from pairs containing S1 and RCM images. However, that was not the case. S1 was paired with S1 and RCM with RCM. Please rephrase for clarity.

We agree. This is clarified in the revised manuscript:

> *"[…], the ECCC-ASITS determines the sea ice motion at a 25 and 6.25 km resolution from S1 or RCM SAR images."*

Line 265: 'will BE assessed': This is corrected in the revised manuscript

Major comment on the dataset

The dataset's structure is such that start/end coordinates of all three nodes of an element are provided for each deformation value. Therefore, the coordinates are duplicated for two neighbour elements. Therefore, the same coordinates are provided (duplicated) as many times as a node is shared by neighbour elements. On average, the dataset size is three -- four times larger than it should be. Moreover, this structure is not convenient to use for plotting and analysis.

[…] I propose to use a more efficient data structure instead. […] Detailed explanations are provided in the notebook (see also the corresponding pull request to the McGill-sea-ice/SIDRRpy repository).

We are very grateful for this comment. We originally favored using the same dimension (n :: number of triangle) in the netcdf files for all variables, meaning that each vertex position is registered multiple times. As you point out, it has no effect other than on the size of the dataset.

We find your modification to be very useful, not only reducing the size of the dataset, but also simplifying the analysis scripts (e.g., we no longer need to reconstruct the original lat-lon vector).

We therefore applied your suggestion and modified the revised dataset as follows:

- We no longer list the vertex coordinates, and instead list the tracked features' start and end positions only once (i.e., 4 variables: start_lat, start_lon, end_lat and end_lon, instead of the original 12).
- A new dimension (npts) is included to represent the number of triangulated tracked features.
- A new variable is added (pts_idpair) to associate the tracked points to their corresponding SAR image pair ID.

The manuscript (Table 1 and section 4.1, L150-155) is also revised to reflect these changes. Note that there are no other modifications and the SIDRR data is otherwise exactly the same as the previous version.

These modifications reduced the size of the dataset from ~25Gb (compressed to 10Gb on Zenodo), to ~15Gb. The new version of the SIDRR dataset will be updated with a new DOI.

**Answers to essd-2024-227 RC2**

October 9, 2024

Note:

- The referee comments are shown in black,
- The authors answers are shown in blue

The manuscript is concerned with a dataset of sea ice and rotation rates (SIDRR) based on Sentinel-1 and RadarSAT SAR imagery is presented. The method used to build the dataset is very well described, along with a budget of the errors the estimated parameters can be affected.

This is an excellent work, the product it describes waits only to be exploited by the sea ice community.

I do not have specific comments, so I recommend publication in its present form.

We thank the reviewer for their positive review.

**Answers to essd-2024-227 RC3**

October 9, 2024

Note:

- The referee comments are shown in black,
- The authors answers are shown in blue,
- *Quoted texts from the revised manuscript are shown in italic and in dark blue.*

This study presents a Sea Ice Deformation and Rotation Rates (SIDRR) dataset, derived from sea ice motion vectors obtained through the Environment and Climate Change Canada automated sea ice tracking system (ECCC-ASITS). The dataset spans from 2017 to 2023 and covers the pan-Arctic region, using Synthetic Aperture Radar (SAR) imagery from the Sentinel-1 (S1) and RADARSAT Constellation Mission (RCM) satellites. The paper offers valuable insights into the processing methods and analysis of sea ice deformations across multiple spatial and temporal scales, and serves as an important contribution to sea ice dynamics research.

The overall structure of the manuscript is sound, and the scientific value of the dataset is well articulated. The data processing methods are thorough, and the analysis of uncertainties is a crucial and valuable aspect of the study. That said, there are several points where the paper could be improved. Below are specific suggestions and observations.

We thank the reviewer for their useful review. We agree and address their comments below.

Specific comments:

1.  Line 40: The introduction of the RADARSAT Geophysical Processing System (RGPS) dataset lacks a clear transition and purpose. To improve clarity, it would be beneficial to emphasize why the RGPS dataset is insufficient for certain studies, particularly those requiring higher spatial and temporal resolution to capture fine-scale sea ice deformation. This would provide a stronger foundation for introducing the new dataset developed in the paper.

The RGPS dataset is introduced here as it is seminal for the study of sea ice dynamics from SAR trajectories. We note here that the RGPS dataset is not necessarily limited (compared to the current SIDRR dataset) in terms of the resolution or its use to define multi-scale properties: the

SIDRR dataset has similar limitations and even higher uncertainties. The limitation of the RGPS dataset rather lies in its ending in 2008. This is a limitation for forecast systems, as validation against RGPS requires the production of hindcast simulations. This is clarified in the revised manuscript at L53-54:

> *"This widening gap furthermore complicates the validation of sea ice dynamics in forecast systems, requiring hindcast simulations that are increasingly different from their operational framework."*

2.   Line 74: You should explicitly mention the spatial resolution of both products. The temporal resolution differences between S1 and RCM are also crucial for the analysis. S1 might have less frequent passes (especially due to the polar orbit), whereas RCM can provide higher temporal resolution with more frequent observations.

We added some details at L79-82 in the revised manuscript, as indicated below. Note that the temporal resolution is a bit tricky to discuss as the coverages evolve over time. Both the S1 and RCM satellites have a similar range of pass frequency (from sub-daily to 3 days), although with different regional distributions.

> *"Before the loss of S1B in December, 2021, S1 had a slightly higher pass frequency in the central Arctic, but the RCM had a better coverage of the Canadian Arctic (see Howell et al., 2022, for more details). After 2021, the three-satellite RCM has much better coverage than S1A alone. Currently, RCM provides excellent daily or sub-daily coverage over the Canadian Arctic, and about once per three days over the Eurasian part of the Arctic. The original resolution of the S1 and RCM images depend on the beam mode but are all resampled to 200 m prior to running the ASITS."*

3.   Line 81: Explain the rationale for the 30% overlap to help the reader understand its importance. You could clarify whether this overlap increases the confidence in feature tracking or provides redundancy to account for data gaps.

This overlap threshold has no impact on the feature tracking and only serves to optimize the amount of data processed within an operational framework. This is mentioned at L80. This pairing method causes some gaps in data coverage, as shown in Fig. 5 and discussed at L196-199.

4.   Line 86: Define what is meant by "raw" data here. Is the goal to maintain the dataset's flexibility so that users can choose their own post-processing methods, or is there another reason to keep the data in this format?

We were using "raw" here to designate the outputs from the ASITS, as opposed to using the aggregated sea ice motion products gridded at 6.25 km and 25 km resolutions. This choice is indeed meant to keep the dataset as flexible as possible, but also to conserve the Lagrangian character of the tracked motion. This is now specified at L92-94 in the revised manuscript:

*"Here, these ASITS sea ice motion vector outputs are used directly to generate the new sea ice deformation product, keeping their Lagrangian character (i.e., we do not use the ECCC-ASITS SIM gridded datasets, which are produced by aggregating and post-processing the SIM vectors)."*

5.   Line 201: "Validation of the SIDRR features is complicated by 1. the multi-scale nature of sea ice deformations...", the first point regarding the multi-scale nature could benefit from additional explanation. What exactly about the multi-scale nature complicates validation? Adding a sentence to explain how multi-scale deformations make it difficult to compare or validate the data would be helpful.

We are referring here to the fact that the magnitude of the sea ice deformation is scale dependent (both spatially and temporally). This statement is clarified in the revised manuscript:

*"Validation of the SIDRR features is complicated by 1. the scale dependency of the computed sea ice deformation magnitude, making for difficult interpretation of side-by-side comparisons unless the products are computed at the exact same spatio-temporal scale"*